# A hemispheric two-channel code accounts for binaural unmasking in humans

Jörg Encke [1,2]✉ & Mathias Dietz [1,2]

Sound in noise is better detected or understood if target and masking sources originate from different locations. Mammalian physiology suggests that the neurocomputational process that underlies this binaural unmasking is based on two hemispheric channels that encode interaural differences in their relative neuronal activity. Here, we introduce a mathematical formulation of the two-channel model – the complex-valued correlation coefficient. We show that this formulation quantifies the amount of temporal fluctuations in interaural differences, which we suggest underlie binaural unmasking. We applied this model to an extensive library of psychoacoustic experiments, accounting for 98% of the variance across eight studies. Combining physiological plausibility with its success in explaining behavioral data, the proposed mechanism is a significant step towards a unified understanding of binaural unmasking and the encoding of interaural differences in general.

[1] Department of Medical Physics and Acoustics, University of Oldenburg, Oldenburg, Germany. [2] Cluster of Excellence 'Hearing4all', University of Oldenburg, Oldenburg, Germany. ✉email: joerg.encke@uol.de

The auditory system has the challenging task of restoring the spatial properties of an acoustic scene based solely on the signals arriving at the two ears. A critical source of information in this process is the difference in arrival time between the signals at the two ears. The delay line or Jeffress model[1], one of the longest-standing models of sensory-neuronal computation, suggests that an array of coincidence-detecting neurons compares neuronal signals from the two cochleae. According to this model, each neuron in the array is associated with a best delay $\tau$ compensating for a specific interaural time difference (ITD); the neuron that compensates best for the ITD would then show the strongest response. This concept corresponds to cross-correlation and postulates a place code for ITD. The delay-line concept was supported by the success of quantitative cross-correlation-based models that were able to predict a variety of human psychoacoustic data[2–5]. Equally compelling, the predicted arrangement of axonal delay lines has been found in the nucleus laminaris of the barn owl[6], a spatial hearing specialist. In mammals, however, no such structure has been found. Instead of a nearly frequency-independent distribution of best-delays centered around $\tau = 0$ as ideal for the Jeffress model[7], studies found the best delay of neurons in each hemisphere of the brain to be centered around 1/8th of the cycle duration[8–10] (see the visualization in Fig. 1c). Mammals thus seem to lack the topographical map of ITDs, as postulated by Jeffress. These findings resulted in the formulation of an alternative coding hypothesis: The two-channel model. Instead of the large number of systematically tuned coincidence detectors used by the Jeffress model, this model relies on the activity within only two broad hemispheric channels[8]. Instead of

the Jeffress place code, the ITD is encoded by the relative firing-rate change within both channels. The two-channel code thus represents a rate code. The two-channel approach has been incorporated into several quantitative models dealing with various aspects of binaural hearing[11–14], but between them, these models still lack the predictive power of cross-correlation-based approaches. As a consequence, and despite the apparent lack of systematic delay lines in mammals, Jeffress-type models are still widely used to account for experimental data in humans, especially when dealing with phenomenons beyond sound localization[15,16].

In addition to sound localization, listening with two ears also provides a benefit in complex environments in which a target sound is masked by sounds from another location[17]. This binaural unmasking has been studied extensively using tone-in-noise detection experiments, resulting in a large body of highly reproducible data[3,18–20].

These studies consistently found that tone-in-noise detection improves considerably when an interaural difference is introduced into either the tone or the masker. If the masker is identical in both ears, anti-phasic 500-Hz tones can be detected at a sound level 15 dB lower than for in-phase tones[18]. This benefit is purely binaural: monaural detection thresholds are unaffected by changes in the tone phase, even though the waveform of the noise signal changes depending on the phase of the added tone (see Fig. 1a).

Tone-in-noise detection thresholds depend on several stimulus features, including noise correlation, noise ITD, interaural phase relation of noise and target, and noise bandwidth. Models based on

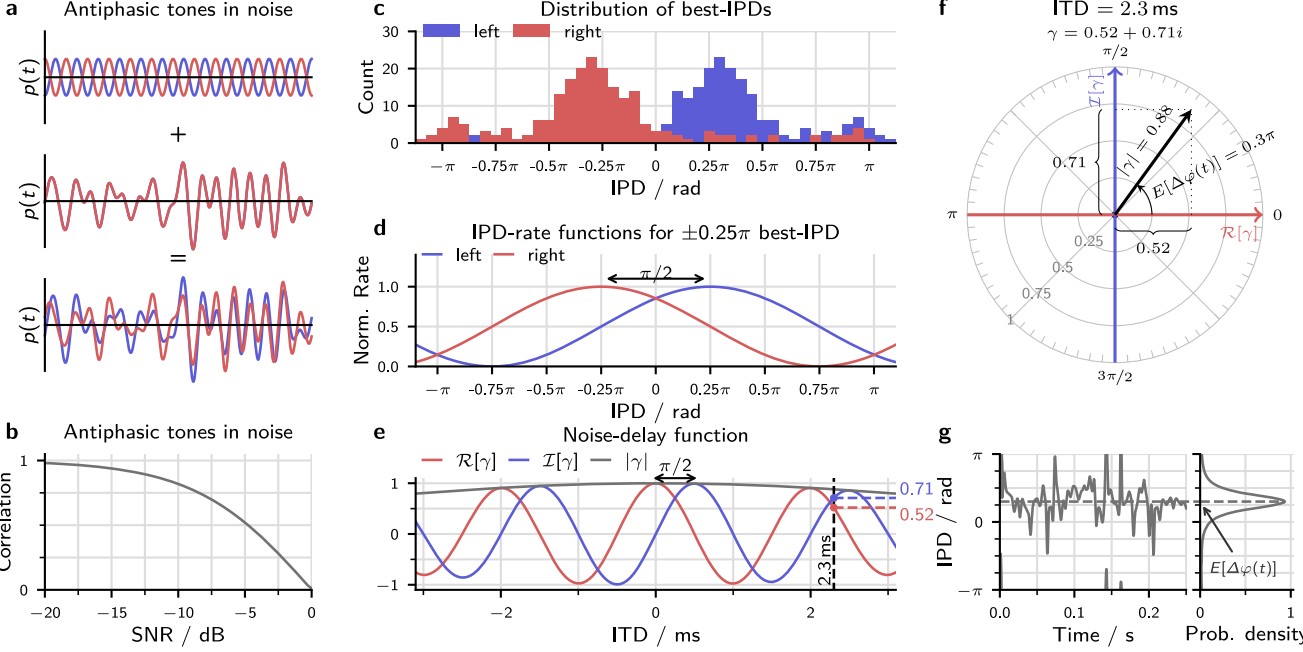

**Fig. 1 Visualization of the modeling approach. a** Schematic visualization of the signals in a tone-in-noise experiment. In this example, anti-phasic tones (top panel) are added to the same noise token (middle panel), resulting in a partly decorrelated stimulus (bottom panel). **b** The amount of decorrelation depends on the level of the added tone. This panel shows the correlation as a function of the signal-to-noise ratio (SNR) calculated for an anti-phasic tone-in-noise as visualized in **a**. **c** Histograms of the best-IPDs of 217 neurons that were digitized from[10]. For visualization, the histogram of the right hemisphere was added as an identical but mirrored version of the left hemisphere. **d** Assumed IPD-rate functions for two neurons with $\pm\pi/4$ best-IPD. **e** Components of the complex-valued correlation coefficient for different ITDs. Results were calculated for Gaussian noise after applying a 500 Hz-centered gammatone filter with 79 Hz equivalent rectangular bandwidth. The real and imaginary parts of $\gamma$ show the same $\pi/2$ phase shift as the IPD-rate functions in **d**. The dashed vertical line indicates an ITD of 2.3 ms with the two vertical lines indicating the associated values of the real and imaginary part. **f** Visualization of the different components of the complex correlation coefficient $\gamma$ in the complex plane for the same noise and the 2.3 ms ITD indicated in **e**. **g** Instantaneous IPD $\Delta\varphi(t)$ for the same noise as used in **e** and **f**. The right subplot shows the corresponding probability density function[26]. The expected value of this distribution equals the argument of the stimulus coherence $E[\Delta\varphi(t)] = \arg(\gamma) = 0.3\pi$.

the cross-correlation function account for these dependencies by detecting changes in the maximum of the cross-correlation function $\rho(\tau)$ which are caused by the tone[2,3]. These models benefit from the large array of differently tuned coincidence detectors which enables them to use the most informative delay element, $\rho(\tau_{\text{best}})$, for the respective task. Models that lack the array of delay-tuned detectors, such as the two-channel model, fall short in terms of both accuracy and comprehensiveness[11,12,21].

Adding a tone with an interaural phase difference (IPD) to a correlated masker not only reduces the correlation but also introduces fluctuations in the stimulus IPD[22,23]. While the reduction in correlation and the fluctuation strength are closely connected, this relation does not always hold. This is illustrated by comparing two cases: (A) two noise tokens from independent sources; (B) two noise tokens from the same source where one token is phase-shifted by $\pi/2$. In both cases, the correlation between the tokens is zero. In the case of the two independent tokens, however, the IPD fluctuates randomly, whereas, by definition, (B) has a constant IPD. The amount of IPD fluctuation thus offers additional information about the underlying binaural statistics and has been proposed as an alternative metric for binaural detection[10,22,24]. Yet, this approach has received relatively little attention in quantitative modeling.

This study aims to remedy the current limitations of the two-channel model in accounting for binaural unmasking experiments. By introducing a new mathematical representation of the two-channel model, we reveal a direct connection to the amount of IPD fluctuation proposed to underlie binaural detection. The proposed representation of the two-channel model also creates the theoretical foundation for a new understanding of IPD encoding in the mammalian brainstem. We evaluate this new approach to the two-channel model by comparing predictions of a signal-detection model against an extensive library of binaural detection experiments.

## Results

**A mathematical representation of the two-channel model.** The two-channel model assumes that ITDs are encoded in the activity within two broad hemispheric channels. These channels are represented by the mean activity of neurons in the left and right brain hemispheres[8]. ITD-sensitive neurons in the midbrain have been found respond strongest to ITDs around 1/8th of the cycle duration[8–10] which is equivalent to an IPD of $\pi/4$. If we assume that each channel is represented by the mean activity of all units within the hemisphere, we can represent them as one correlator each. These correlators show best-IPDs of $\pm\pi/4$. As a consequence, their relative phase difference is $\pi/2$ so that the two channels are orthogonal[10] (see Fig. 1d).

An equivalent but mathematically more convenient method of representing the two $\pm\pi/4$ channels is to use one correlation with zero best-IPD and a second correlation with $\pi/2$ interaural phase offset. This phase offset can be obtained by Hilbert transformation $\mathcal{H}$ of either the left ear signal $l(t)$ or the right ear signal $r(t)$. Both real-valued correlations can then be expressed by a single complex-valued correlation coefficient $\gamma$:

$$\gamma = \frac{<l(t)r(t)>}{\sqrt{<|l(t)|^2><|r(t)|^2>}} + i\frac{<\mathcal{H}[l(t)]r(t)>}{\sqrt{<|l(t)|^2><|r(t)|^2>}}, \quad (1)$$

where $i$ indicates the imaginary unit. Figure 1e shows noise-delay functions for the two correlators (blue and red line). The benefit of using this complex-valued representation is that it is equivalent to directly calculating a single correlation coefficient for the complex-valued or analytic representations of the two signals

$l_a(t) = l(t) + i\,\mathcal{H}[l(t)]$ and $r_a(t) = r(t) + i\,\mathcal{H}[r(t)]$:

$$\gamma = \frac{<l_a^*(t)r_a(t)>}{\sqrt{<|l_a(t)|^2><|r_a(t)|^2>}}, \quad (2)$$

where the asterisk indicates the complex conjugate, and the angular brackets the ensemble average. This expression is also called the complex-valued correlation coefficient. It is well known in other fields of physics that deal with waves such as optics[25,26], allowing to inherit its established properties and advantages.

Figure 1f represents $\gamma$ as an arrow in the complex plane where the $x$- and $y$ axis equal its real and imaginary parts. The example was calculated for an ITD of $\Delta t = 2.3$ ms which is also indicated by the dashed line in Fig. 1d. Instead of using the real and imaginary part, $\gamma$ can also be described by the angle (argument arg($\gamma$)) and the length (modulus $|\gamma|$) of the arrow. These two values have some interesting properties. The argument arg($\gamma$) equals the expected value of the distribution of instantaneous IPDs, or in other words, the time-averaged IPD (see Fig. 1g). The modulus $|\gamma|$ is a measure for the consistency of left and right instantaneous phases and thus for the IPD-fluctuation strength. We will refer to $|\gamma|$ as the interaural coherence[10].

Note that there are several differing definitions of coherence. Our use of coherence as $|\gamma|$ is a typical time-domain definition[25]. In general signal processing, the coherence function is instead defined in the frequency domain and calculated as the normalized absolute value of the cross-spectral power density (CSPD)[27]. The two definitions are closely related, as the time-domain coherence can also be defined by using a Fourier transform of the CSPD (see "Methods" for details). In binaural research, a third definition exists, where interaural coherence is sometimes used to refer to the maximum of the real-valued cross-correlation function[28]. It is similar but not equivalent to the more general definitions.

The derivation of $\gamma$ as a mathematical representation of the two-channel model highlights two essential properties of the model: Firstly, the two-channel model can act as a perfect encoder for IPDs, and secondly, the two-channel model also encodes information about the amount of fluctuation in IPD. If these fluctuations can indeed be used for explaining binaural unmasking, as proposed in refs. [10,22,24], then this should also be possible by using $\gamma$. The following section will thus develop a signal-detection model to predict binaural detection based on the quantity $\gamma$.

**A model of binaural detection.** Tone-in-noise detection is usually performed as an alternative forced-choice task with one or more reference intervals containing only the reference noise and a target interval in which the tone is added to the noise. These studies aim to determine the signal-to-noise ratio (SNR) at which the subject can identify the target stimulus with a predefined sensitivity.

A computational model was used to test the hypothesis that binaural tone-in-noise detection can be explained based on the complex correlation coefficient $\gamma$. In the model, threshold SNRs are calculated directly from the absolute difference between the complex correlation coefficients of the masker and target stimuli. Correlation coefficients were calculated based on the spectral properties of the two input stimuli, assuming only a peripheral bandpass filter. In addition to this binaural detection path, a monaural pathway provides an SNR-based detection cue in stimuli with few or no binaural cues. A mathematical description of the model is given in "Methods".

Four stimulus parameters were necessary to define the stimuli used in this study: The IPD of the noise as a function of angular frequency $\Delta\varphi(\omega)$, the noise correlation $\rho_N$, the IPD of the target tone $\Delta\phi$, and the noise bandwidth $\Delta\omega$. For example, for an out-of-phase tone in 900 Hz wide, correlated noise with 2.3 ms ITD (as used in Fig. 1e–g), the parameters would be set to

**Table 1 Aggregation of stimulus and model parameters.**

| Study | Stimulus parameter | | | | Model parameter | | | |
|---|---|---|---|---|---|---|---|---|
| | $\Delta\varphi(\omega)$ | $\rho_N$ | $\Delta\psi$ | $\Delta\omega/2\pi$ | $\hat\rho$ | $\sigma_{bin}$ | $\sigma_{mon}$ | $R^2$ |
| Pollack & Trittipoe[29] | 0 | $\rho + \Delta\rho$ | / | 1 kHz | 0.92 | 0.42 | / | 0.97 |
| Robinson & Jeffress[30] | 0 | −1 to 1 | $0, \pi$ | 0.9 kHz | 0.92 | 0.31 | 0.76 | 0.98 |
| Bernstein & Trahiotis[16] | 0 | −1 to 1 | $\pi$ | 25 Hz to 0.9 kHz | 0.97 | 0.54 | 0.76 | 0.97 |
| Langford & Jeffress[19] | $\omega\Delta t$ | 1 | $0, \pi$ | 0.9 kHz | 0.95 | 0.33 | 0.70 | 0.96 |
| van der Heijden & Trahiotis, 1999[3] | $\omega\Delta t$ | 1 | $0, \pi$ | 0.9 kHz | 0.90 | 0.19 | 0.61 | 0.95 |
| Rabiner et al.[20] | $(\omega - \omega_0)\Delta t$ | 1 | $\pi$ | 1.1 kHz | 0.85 | 0.24 | 0.71 | 0.95 |
| Bernstein & Trahiotis[15] | $\omega\Delta t$ | 0.498 to 1 | $\omega_0\Delta t + \pi$ | 0.1 kHz, 0.9 kHz | 0.89 | 0.52 | 0.93 | 0.96 |
| van de Par & Kohlrausch[32] | $0, \pi$ | 1 | $0, \pi$ | 5 Hz to 1 kHz | 0.97 | 0.38 | 0.76 | 0.91 |
| Individual parameters optimized for each experiment: | | | | | | | | 0.98 |
| One parameter set optimized for all experiments: | | | | | 0.96 | 0.40 | 0.74 | 0.93 |

Parameters used to simulate experiments from eight different studies. The last column states the resulting coefficient of determination $R^2$, which can be interpreted as the proportion of the variance in the dataset that is explained by the model. The last row lists a single set of parameters that were optimized to minimize $R^2$ for all experiments.

$\Delta\varphi(\omega) = \omega \times 2.3$ ms, $\rho_N = 1$, $\Delta\psi = \pi$, and $\Delta\omega = 2\pi \times 900$ Hz. Table 1 summarizes the stimulus parameters of all experiments that are discussed below. Three parameters define the model itself: parameters $\sigma_{bin}$ and $\sigma_{mon}$, which directly determine the binaural and the monaural detection sensitivity, and a third parameter $\hat\rho$, which limits the maximal sensitivity to changes in coherence. Model parameters were optimized separately for each experiment because detection thresholds for identical stimuli were not always identical across studies. This finding is unsurprising since most experiments were conducted with few subjects. Table 1 summarizes the resulting model parameters.

**Simulated datasets**. The first dataset by Pollack & Trittipoe[29] is not from a tone-in-noise experiment but directly quantified the sensitivity to changes in coherence. For the model, this sensitivity was directly calculated using Eq. (8). Experimental data and model results are shown in Fig. 2a.

In the next two experiments, the reference also consisted of noise with predefined interaural correlations. However, the target correlation was not manipulated directly but was changed by adding a tone to the partly correlated noise. Robinson & Jeffress[30] collected results for both in-phase and anti-phasic tones (see Fig. 2b). The experiment with anti-phasic tones was repeated by Bernstein & Trahiotis[16], who also collected data at different noise bandwidths (see Fig. 2c). The change in coherence that arises from adding a tone at a given SNR depends on the initial noise correlation $\rho_N$ and on the difference between the tone IPD and the noise IPD[31]. The coherence change is greatest when the two IPDs are out of phase, while there is no influence when the IPDs are the same. In Fig. 2b, this is reflected in the large difference between the threshold SNRs of the in-phase and anti-phasic conditions when $\rho_N = -1$ and $\rho_N = 1$. The improvement in threshold SNR with increasing bandwidth, as seen in Fig. 2c, can be explained solely by the filter property of the auditory periphery. Only noise energy that falls within the peripheral filter interacts with the tone, thus determining the coherence. This peripheral filtering improves the SNR and, thus also the nominal threshold SNR. For large stimulus bandwidths far exceeding the peripheral bandwidth, this improvement equals 3 dB/octave.

Instead of directly changing the masking noise correlation, the next set of studies by Langford & Jeffress[19], and van der Heijden & Trahiotis[3] applied an ITD to the noise before adding the target tone (Fig. 2d, e). The added ITD results in both a reduction in coherence $|\gamma|$ and a shift in the noise IPD (see Fig. 1e, f for an example). The change in noise IPD results in periodic oscillations of the threshold SNR, as the effectiveness of the added phasic or

anti-phasic tone changes with noise IPD. The periodic oscillation is superimposed by an overall increase in threshold SNR with ITD. Rabiner et al.[20] conducted a slightly different but related experiment: instead of applying the ITD to the whole noise, it was applied only to the envelope of the masking noise, which keeps the noise IPD fixed at zero (Fig. 2f). This removes the oscillations in the threshold SNR while resulting in the same increase in threshold SNR as when using the regular ITD.

Yet another stimulus variation was introduced by Bernstein & Trahiotis[15] who added anti-phasic tones to noises of different interaural correlations and applied the ITD to the whole signal. This modification keeps the phase relation between noise IPD and tone IPD fixed at $\pi$ so that the ITD only influences the stimulus coherence (Fig. 2g). As in the study in ref. [20], this results in an increase in the threshold SNR without oscillations. The increase is less pronounced at low noise correlations because the effect of the ITD on coherence diminishes with decreasing noise correlation.

Figure 2h shows experimental results from van de Par & Kohlrausch[32] as well as simulated threshold SNRs for a large range of bandwidths from 5 Hz to 1 kHz in two configurations with binaural cues and one configuration with monaural cues only. The model accounts for the bandwidth dependence of detection thresholds because of its bandpass filter. The filter does not considerably affect the noise energy at very low bandwidths, so the threshold SNR remains constant. The predicted threshold SNR improves by 3 dB per octave at large bandwidths.

With parameters that were optimized individually for each experiment, the model could account for 91 to 98% of the respective variance (see Table 1). For all datasets together, keeping the individual parameters, the model accounted for 98% of the total variance. Figure 2i visualizes this high correlation between modeled and experimental thresholds by plotting one against the other. A single set of parameters, optimized to reduce the variance across all datasets, still accounted for 93% of the total variance, despite the deviations in the experimental threshold for identical stimulus parameters mentioned above.

## Discussion

The complex-valued correlation coefficient model proposed here accounted for nearly all aspects of the psychophysical datasets examined in this study, with limitations discussed below. The modeled binaural sensitivity is directly proportional to the difference in the z-transformed complex correlation coefficient $\gamma$ between target and reference. The bandpass filter is the only pre-processing stage necessary to account for these datasets. The following discussion will thus focus on these two components of the model.

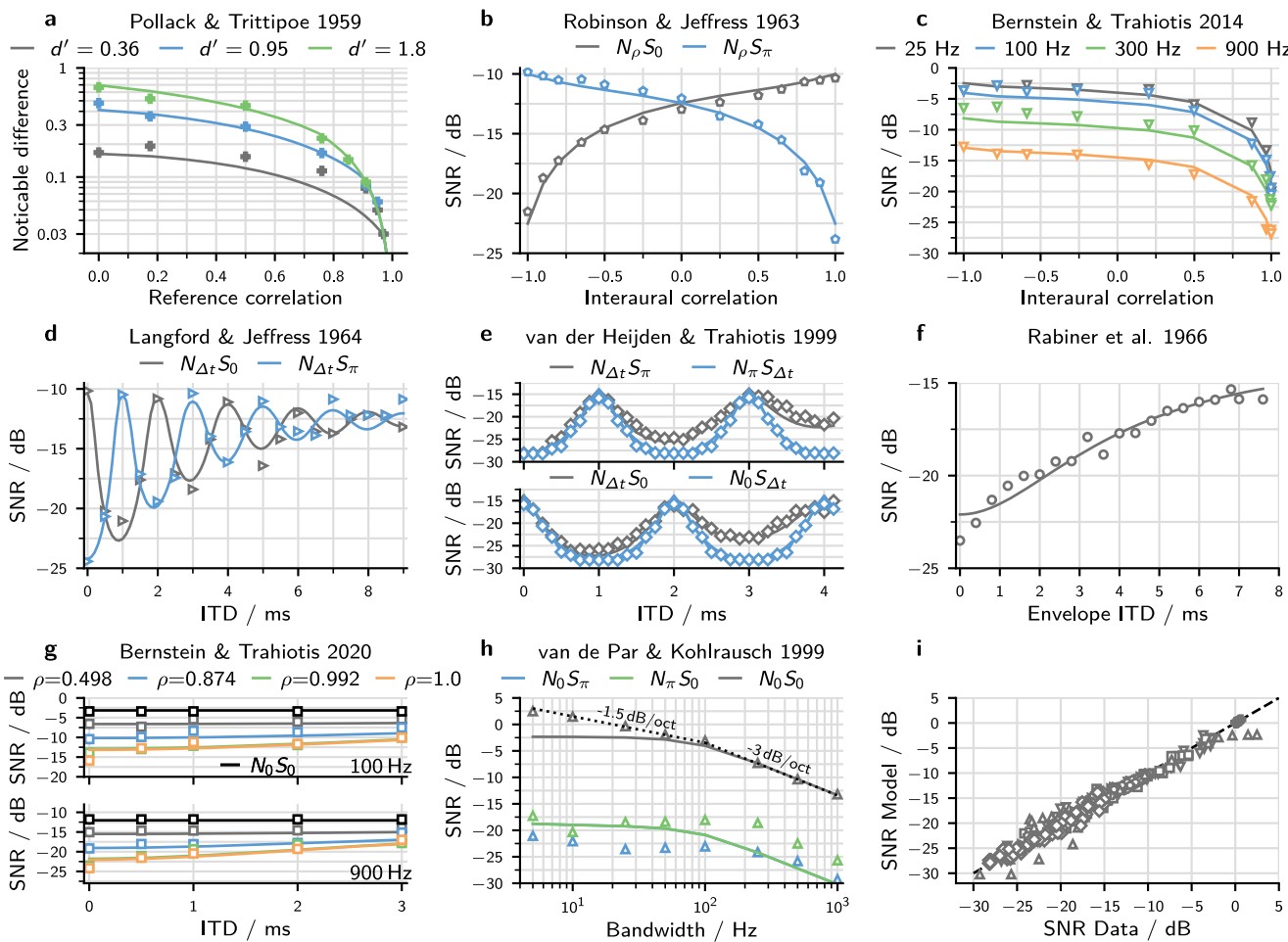

**Fig. 2 Experimental data and model results for all experiments.** In all cases, symbols indicate the experimental data as digitized from the respective study. Simulated thresholds are shown as lines in matching colors. **a** Incoherence detection thresholds. The reference noise correlation serves as abscissa and the required change in correlation as ordinate. The different colors indicate different sensitivity indices, i.e., different threshold definitions. **b**–**h** Threshold SNRs for seven different studies with the threshold defined as in the respective study. Colors differentiate between conditions. **b** Dependence on noise correlation. Colors differentiate between the use of in-phase ($N_\rho S_0$) or anti-phasic ($N_\rho S_\pi$) tones. **c** The same experiment as in **b** for anti-phasic tones only. Colors mark different stimulus bandwidths. **d** Dependence on noise ITD. Colors indicate the use of an in-phase (gray) or anti-phasic (blue) target tone. **e** The same experiment as in **d** with higher ITD resolution (gray data). Additional conditions depict thresholds when the ITD was applied to the tone instead of the noise (blue). **f** Similar to **d** but instead of a regular ITD, the ITD is only applied to the envelope of the noise. **g** Similar to **d** but the ITD is applied to the whole stimulus (noise and tone). Colors indicate different noise correlations while the ($N_0S_0$) condition is shown in black. Data for a 100 Hz masker bandwidth is shown on the top and 900 Hz on the bottom. Results for the $\rho = 0.992$ condition are partly concealed by those of the $\rho = 1$ condition. **h** Dependence on stimulus bandwidth. Colors mark the interaural configuration, including one without binaural cues (gray). Model results for the $\mathbf{N_0S_\pi}$ condition are concealed by those for the $\mathbf{N_\pi S_0}$) condition. **i** Summarizing scatter plot of all 329 simulated threshold SNRs plotted against their experimental counterpart. The dashed diagonal indicates points of equality. Symbols correspond to the respective symbols from **b**–**h**.

In the first three experiments (Fig. 2a–c), no phase or time shifts were added to the signal. Consequently, the imaginary part of $\gamma$ was always 0 so that $\gamma$ equaled the real-valued correlation coefficient $\rho$. This means that the model would show identical results if it were based solely on $\rho$. The comprehensiveness and accuracy of $\rho$-based models for this kind of stimuli have been demonstrated previously[2].

The experiments shown in Fig. 2d–i, included ITDs or IPDs, so that $\gamma$ was generally not real-valued. In these cases, models based on real-valued correlation alone need to include larger parts of the cross-correlation function $\rho(\tau)$[2,3]. Alternatively, as shown in this study, this kind of data can be explained entirely by the complex correlation coefficient. To better understand the underlying mechanism, Fig. 3a visualizes the z-transformed coefficients for threshold SNRs for the data of ref. [3]. The illustrated complex space can be interpreted as a binaural feature space, where the distance between the reference and target is directly proportional to the

binaural sensitivity $d'_{\text{bin}}$. It is apparent that this distance is determined by both the coherence, reflected in the length of the vector, and the mean IPD reflected in its angle. The relative contributions of length and angle depend on the specific stimulus condition. For conditions where the difference between the mean IPD of the noise and the tone equals 0 or $\pi$, the added tone cannot influence the mean IPD of the resulting signal. In these cases, binaural detection was based solely on a change in coherence. If the difference is $\pi$, the coherence change caused by the target is large, so the binaural cue is generally large as well. When the difference is 0, detection relied mostly on monaural cues, as visible in Fig. 3a: the vectors of reference and target are nearly equal at $\Delta t = 1$ ms. In this situation, the availability of binaural cues increases with decreasing noise coherence, as the added tone can then increase the coherence of the target relative to the reference (see $\Delta t = 3$ ms in Fig. 3b). The decrease in coherence with ITD is visible in the decreasing length of the reference vectors.

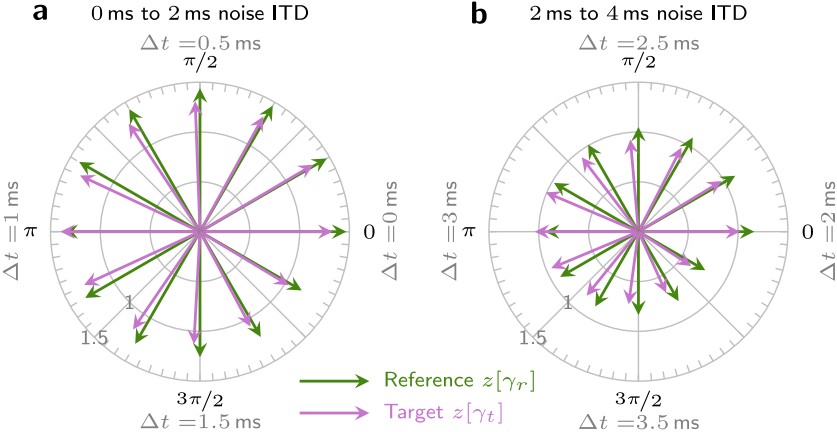

**Fig. 3 Vector visualization of the z-transformed correlation coefficients.** The figure shows results for selected stimuli of the $\mathbf{N}_{\Delta t}\mathbf{S}_\pi$) condition of van der Heijden & Trahiotis[3] (also see Fig. 2). Results for reference stimuli are shown in green while target stimuli are shown in purple. The length of each vector represents the z-transformed coherence of the respective stimulus, while the angle represents the mean IPD. Target stimuli, shown in purple, contain both tone and noise at the experimental threshold SNR. Noise-ITDs in the range of 0–2 ms are shown in panel **a** and 2–4 ms in panel **b**. Changes in $\Delta t$ are directly reflected by the average noise IPD, which determines the polar angle of the reference, so polar-labels state both the angle and the respective noise ITD. The distance between the tips of the reference and target vectors is directly proportional to the strength of the binaural cue.

Of the data simulated in the present study, only that of van der Heijden & Trahiotis[3] and Langford & Jeffress[19] include stimuli with mean IPD differences between noise and tone other than 0 or $\pi$. Only these intermediate differences cause a change in both coherence and angle of the target stimulus (Fig. 3). In these intermediate cases, it is particularly advantageous to use the complex plane of the z-transformed correlation coefficient $z[\gamma]$ as a two-dimensional acoustic stimulus feature space. A key finding of the present study is that the acoustic feature space can be used directly as a perceptual feature space so that the distance between two stimuli in this space is proportional to the binaural sensitivity index $d'_{\text{bin}}$. The complex plane of $z[\gamma]$ can thus be interpreted as a perceptually uniform space like, for example, the CIELAB color space that is commonly used to represent color difference sensitivity[33].

If $z[\gamma]$ is interpreted as a perceptually uniform space, it should be possible to use the same space to explain related phenomena, such as ITD discrimination. For tones, ITDs are equivalent to IPDs. Since IPDs are reflected in the argument of $\gamma$, IPD discrimination sensitivity can be directly derived from Eq. (8):

$$\Delta\sigma_{thr} = 2\arcsin\left(\frac{d'_{\text{bin}}\sigma_{\text{bin}}}{2\operatorname{arctanh}(\hat{\rho})}\right). \tag{3}$$

Using the set of parameters summarized in Table 1. with $d'_{\text{bin}} = 1$ resulted in IPD thresholds equivalent to ITDs in the range of 41 µs to 117 µs (median 60 µs); this is within the range of experimentally obtained thresholds at 500 Hz. For discrimination around zero ITD, experimental thresholds are on average a little lower[34], but for discrimination around $\pi$ thresholds are above the model median[35].

As elaborated above, the proposed measure $\gamma$ is equivalent to calculating two normalized correlation coefficients. Naturally, this assumption implies the existence of some form of neuronal normalization process, an assumption that is shared with the majority of the cross-correlation-based models. Mathematically, this normalization could take place using monaural information such as the activity of the auditory nerve[13]. It has, however, been noted that this process would have to be extremely precise[36]. Alternatively, normalization could also be based on the activity in "anti-coincidence" detecting neurons such as those found in the lateral superior olive[37]. The firing rate in these neurons behaves inversely to those that act as coincident detectors and increases

with decreasing correlation. Comparison between anti-coincidence and coincidence-detecting neurons could thus be used for normalization. A third method would be to directly use the time course of IPD fluctuation. Instead of encoding the real and imaginary part of $\gamma$, this approach would encode the time-averaged IPD and the coherence. The ability of the auditory system to encode for the former is well established[38]. To encode information equivalent to coherence, the auditory system could directly rely on the amount the IPD fluctuates around its mean. This IPD-fluctuation code would have the benefit of only requiring information about a single quantity—the IPD.

The neuronal substrate necessary to extract $\gamma$ based on the two channels would depend on the underlying mechanism. Representing $\gamma$ directly based on the values of coherence and mean IPD would require a long-term integration mechanism with two subsequent processing stages implementing neuronal equivalents to calculating modulus and argument. If $\gamma$ would instead be represented indirectly via IPD fluctuations, neurons would have to be fast enough to follow these fluctuations. Neurons that do just this have indeed been described: IPD-sensitive neurons can encode the fast fluctuations by means of fast changes in their response rate[39–41]. This second mechanism would also be in-line with a recent neuroimaging study which reported an elevated cortical load when subjects were presented with low-coherence stimuli[42]. The increased load could result from the brain having to deal with localizing a sound source based on increasingly fluctuating IPDs.

In cases where the noise bandwidth is considerably larger than the peripheral filter bandwidth, the coherence function $|\gamma(\tau)|$ is fully determined by the power spectrum of the peripheral filter (see Eqs. (4) and (5)). By substituting $\tau$ with the ITD, the same function can describe the ITD dependence of $|\gamma|$. In the absence of a delay line, the decline of the binaural benefit with masker ITD is therefore a direct cause of the bandwidth after filtering. Langford & Jeffress were the first to describe this relation, and coarsely estimated that a 100-Hz filter bandwidth explains the ITD dependence of their data (see Fig. 2d)[19]. With more quantitative analysis, and assuming a triangular filter, Rabiner et al.[20] found that their data (see Fig. 2f) was best accounted for by a filter with 85 Hz equivalent rectangular bandwidth (ERB). This value is close to the 79 Hz ERB of the 4th-order gammatone filter that was used in the present study (see "Methods"). The ERB was fixed at 79 Hz to reduce the number of free model parameters. This

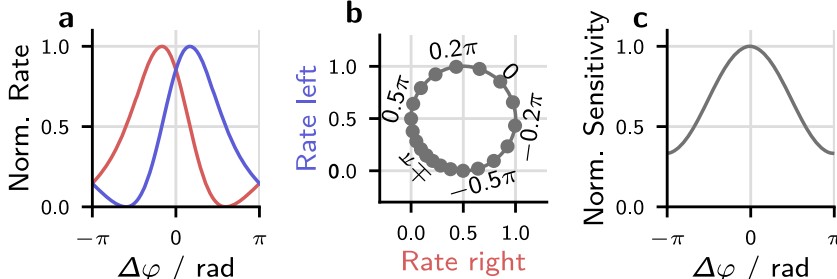

**Fig. 4 Visualization of the impact of non-sinusoidal IPD-rate functions. a** IPD-rate functions that exhibit a steeper slope towards zero IPD. **b** Plotting the firing rates of the left hemisphere against the rates in the right hemisphere results in a plot similar to the complex plane plot shown in Fig. 1g. Markers indicate the location of IPDs evenly spaced by $0.1/\pi$. **c** The uneven spacing of IPDs on the circle shown in **b** results in the sensitivity to changes in IPD to depend on IPD. Shown is the sensitivity normalized to its value at zero IPD.

bandwidth is a typical estimation for the bandwidth of the monaural periphery[43] and has also been employed in other binaural detection models[2,44]. The same filter bandwidth is also responsible for the change in threshold SNR with stimulus bandwidth, as seen in Fig. 2c, where the point at which the simulated threshold SNR starts to decrease is determined primarily by the bandwidth.

While the proposed model was able to account for nearly all characteristics of the datasets shown in Fig. 2, some limitations do remain.

The experiment shown in Fig. 2h revealed differences between the threshold SNRs for in-phase tones in anti-phasic noise ($\mathbf{N}_\pi\mathbf{S}_0$) and anti-phasic tones in in-phase noise ($\mathbf{N}_0\mathbf{S}_\pi$): thresholds in the ($\mathbf{N}_\pi\mathbf{S}_0$) condition were about 4 dB higher than for $\mathbf{N}_0\mathbf{S}_\pi$. The same trend can be seen in the data shown in Fig. 2b and is consistent across other studies[18,30]. This 4 dB difference between the $\mathbf{N}_\pi\mathbf{S}_0$ and the $\mathbf{N}_0\mathbf{S}_\pi$ condition can not be accounted for by the presented model. In the model, the two conditions differ only in their mean IPD (that is, in the argument of the correlation coefficient $\arg(\gamma)$), which is zero for the $\mathbf{N}_0\mathbf{S}_\pi$ condition and $\pi$ for $\mathbf{N}_\pi\mathbf{S}_0$. By using $\gamma$ to predict thresholds directly, the model assumes that the sensitivity to IPDs does not depend on the mean IPD, so predictions for the two conditions are the same. The mean IPD-dependent difference in the experimental thresholds also reflects the previously mentioned difference in sensitivity to changes in IPD, which is lower around $\pi$ than around $0$[35]. Differences in neuronal coding precision can explain both changes in sensitivity. The responses of IPD-sensitive neurons in the auditory brainstem and midbrain usually show their strongest change around IPDs of zero; this has been suggested to facilitate the accurate encoding of IPDs in this region[8,45]. The shape of the IPD-rate functions of these neurons has also been shown to increase IPD sensitivity near zero IPD[46]. The influence of non-sinusoidal IPD-rate functions on the proposed coding mechanism is visualized in Fig. 4. Figure 4a shows asymmetric IPD-rate functions that exhibit a steeper slope towards $\Delta\varphi = 0$ than towards $\Delta\varphi = \pm\pi$. Visualizing these functions in the complex plane (Fig. 4b) results in IPDs that are unevenly distributed with IPDs being more spread out around $0$ than around $\pm\pi$. Consequently, the angle of the complex pointer within this circle will change faster for IPDs around $0$ than around $\pi$. This is also visualized in Fig. 4c, which shows the sensitivity to changes in IPD calculating as the normalized derivative of the pointers angle with respect to IPD. Including these asymmetric IPD-rate functions would result in different sensitivities to changes in IPD as a function of reference-IPD and in $\gamma$ to be less sensitive to IPD fluctuations around $\gamma = \pi$ than around $0$. Differences between $\mathbf{N}_0\mathbf{S}_\pi$ and $\mathbf{N}_\pi\mathbf{S}_0$ are larger at low frequencies and get smaller with increasing frequency[18]. If these differences result from non-sinusoidal IPD-rate functions, then one would also expect the IPD-rate functions to become more

sinusoidal with increasing frequency. This assumption is indeed supported by physiological data[47,48] which shows increasingly harmonic ITD-rate functions with increasing frequency.

Other limitations in explaining experimental data arose from the decision to minimize the model's complexity. In the $\mathbf{N}_0\mathbf{S}_0$ condition shown in Fig. 2h, the model deviates considerably from the data. This deviation results from the sample-to-sample variability of the noise energy, which changes with stimulus bandwidth[32]. This effect cannot be accounted for by the current model implementation, which is based on infinitely long signals. However, a numeric implementation based on finite signal waveforms should account for this phenomenon. Another dataset that has historically been difficult to account for, employs binaural unmasking in reproducible noise token[49,50]. These experiments tested tone-in-noise detection using the same reproducible token across subjects. Like other models based on the average stimulus, we expect the presented model only to explain the average performance but not the variability between individual tokens.

The current model also neglects peripheral processing apart from bandpass filtering. Without this pre-processing, the model cannot account for effects that are associated with the periphery[51–53]. The lack of realistic peripheral processing also limits the ability of the model to account for differences in binaural detection with frequency. IPD and thus ITD sensitivity rely on the auditory nerves' activity to lock onto the stimulus phase (so-called phase-locking). Phase locking declines with increasing frequency[54] which is one of the reasons why sensitivity to ITDs and binaural detection thresholds worsen with increasing frequency[18,34]. The loss of phase-locking could be implemented by a low-pass filter included in a model of peripheral transduction. This filter would remove phase information at higher frequencies while keeping the waveform envelope. Extending the current model with a more detailed periphery should thus help to account for the effects associated with the periphery and the reduction in phase sensitivity at higher frequencies. The current implementation also uses only one single frequency channel. Other experiments, however, such as those employing spectrally complex maskers or maskers constructed from two noise sources with different ITDs, might require a multi-channel implementation of the model. The success of this kind of model extension has recently been demonstrated[55].

Our goal was to test whether the two-channel model that was proposed to underlie ITD sensitivity in mammals could also be used to account for binaural unmasking. By introducing a new mathematical representation of the two-channel model, the complex correlation coefficient, we revealed a direct connection to the amount of IPD fluctuation previously proposed to underlie binaural detection. Using a computational model, we demonstrated that the complex correlation coefficient, and thus the two-channel model, is indeed able to accurately account for many

central aspects of tone-in-noise detection. Compared to the previously best-performing model class, the Jeffress model, our approach is better in line with mammalian physiologic data and represents a considerable simplification as well as a reduction of degrees of freedom.

## Methods

The following will introduce the underlying mathematics that were used for the model implementation. A Python implementation of the model, as well as the scripts for deriving and plotting predictions for all experiments, are openly available[56].

### Deriving the complex correlation coefficient for tone-in-noise detection experiments.

Throughout this study, the complex correlation coefficient was calculated from the cross-spectral power density (CSPD) $S(\omega)$ of the signals. Following the Wiener–Khinchine theorem, the cross-correlation function of two signals is equivalent to the inverse Fourier transform of their CSPD[57]. Consequently, the normalized complex correlation coefficient can be calculated as:

$$\gamma = \frac{\int_{-\infty}^{\infty} S(\omega)\mathrm{d}\omega}{\sqrt{\int_{-\infty}^{\infty} |S_{ll}(\omega)|\mathrm{d}\omega \ \int_{-\infty}^{\infty} |S_{rr}(\omega)|\mathrm{d}\omega}}. \tag{4}$$

Where $S_{ll}(\omega)$ and $S_{rr}(\omega)$ are the power spectral densities of $l(t)$ and $r(t)$. This CSPD-based approach has the benefit of directly resulting in the expected value of $\gamma$ as opposed to a waveform-based implementation where the coherence would have to be estimated as the mean of several instances of the signal waveform.

Given the analytical representations of the left and right signals $l(t)$ and $r(t)$, the effective CSPD $S(\omega)$ was composed of two parts: the CSPD $S_{lr}$ of $l(t)$, $r(t)$ and a transfer function $H(\omega)$ used to account for the bandpass properties of the auditory periphery:

$$S(\omega) = S_{lr}|H(\omega)|^2. \tag{5}$$

The CSPD $S_{lr}$ is directly determined by the stimulus used in the respective experiment and can be formalized as:

$$S_{lr}(\omega) = \begin{cases} \frac{\rho_N}{\Delta\omega} e^{i\Delta\varphi(\omega)}, & \omega_0 - \frac{\Delta\omega}{2} \le \omega \le \omega_0 + \frac{\Delta\omega}{2} \\ \frac{\rho_N}{\Delta\omega} e^{i\Delta\varphi(\omega_0)} + \mathrm{SNR}e^{i\Delta\psi}, & \omega = \omega_0 \\ 0, & \text{otherwise} \end{cases} \tag{6}$$

where $\Delta\omega$ is the bandwidth of a rectangular noise band centered around $\omega_0$ which, in all cases was set to $\omega_0/2\pi = 500$ Hz. $\Delta\varphi(\omega)$ is the IPD spectrum of the noise while $\Delta\psi$ is the IPD of the tone. Both were set according to the conditions used in the respective experiment, as summarized in Table 1. For tone-in-noise detection experiments, $\gamma$ is independent of the absolute level and only depends on the SNR. Consequently, the noise energy was set to one, so that the energy of the tone equals the SNR. Some experiments also made use of noises with different interaural correlations $\rho_N$. The CSPD only contains interaurally coherent energy so that, in these cases, the CSPD of the noise is scaled by $\rho_N$. Assuming the power spectrum of a gammatone filters to account for the bandpass characteristics of the auditory periphery, $|H(\omega)|^2$ was approximated by:

$$|H(\omega)|^2 = \left[1 + \frac{(\omega - \omega_0)^2}{4\pi^2 b}\right]^{-n}, b = \frac{\mathrm{ERB}(n-1)!^2}{\pi(2n-2)! \ 2^{(2-2n)}} \tag{7}$$

where $n$ is the order of the filter and ERB its equivalent rectangular bandwidth[58]. In this study, the filter was centered at 500 Hz with the filter order set to $n = 4$ and the ERB to 79 Hz.

### Modeling the detection performance.

A signal-detection model with two branches, one binaural and one monaural, was used to derive tone-in-noise detection thresholds. The first branch calculates the binaural sensitivity index $d'_{\mathrm{bin}}$ based on the difference of the complex correlation coefficients $\gamma_r$ of a reference signal and of a target signal $\gamma_t$:

$$d'_{\mathrm{bin}} = \frac{|z[\hat{\rho}\,\gamma_r] - z[\hat{\rho}\,\gamma_t]|}{\sigma_{\mathrm{bin}}}. \tag{8}$$

Here, $z[\bullet]$ symbolizes the Fisher's z-transform applied to the modulus of the input while leaving the argument unchanged. This transform normalizes the sampling distribution of the coherence[59]. Direct use of this transformation would result in infinite sensitivity to changes from a coherence of one so that the model parameter $\hat{\rho}$ was introduced. Functionally, this is equivalent to adding uncorrelated noise to the two input signals to account for processing errors on the auditory pathway[60]. The sensitivity of the binaural path is adjusted by the model parameter $\sigma_{\mathrm{bin}}$.

The monaural branch offers sensitivity to increases in stimulus energy between reference and target. As the power of the noise is held constant, this increase is directly proportional to the SNR so that the monaural sensitivity index $d'_{\mathrm{mon}}$ is calculated as:

$$d'_{\mathrm{mon}} = \frac{\mathrm{SNR}_{\mathrm{eff}}}{\sigma_{\mathrm{mon}}}, \tag{9}$$

where $\mathrm{SNR}_{\mathrm{eff}}$ is the effective SNR after peripheral bandpass filtering and $\sigma_{\mathrm{mon}}$ is used to adjust the sensitivity of the monaural pathway.

Assuming a linear independent combination of the monaural and the binaural information, the monaural and binaural sensitivity indices are then combined to the overall sensitivity index:

$$d' = \sqrt{d'^2_{\mathrm{bin}} + d'^2_{\mathrm{mon}}}. \tag{10}$$

All signals that were used in this study are defined by the noise-IPD-spectrum $\Delta\varphi(\omega)$, the tone IPD $\Delta\psi$, the noise correlation $\rho_N$ and the noise bandwidth $\Delta\omega$. With these parameters defined, $d'_{\mathrm{mon}}$ and $d'_{\mathrm{bin}}$ and thus $d'$ only depend on the SNR. Finding the SNR that results in the $d'$ value corresponding to the experimentally defined threshold was solved by using the SLSQP minimization algorithm as implemented in the Python package scipy[61].

### Statistics and reproducibility.

The model performance in accounting for experimental data was evaluated using the coefficient of determination $R^2$ calculated as:

$$R^2 = 1 - \frac{\sum_i (y_i - f_i)^2}{\sum_i (y_i - \bar{y})^2}, \tag{11}$$

where $y_i$ is the $i$th experimental data point and $f_i$ the associated model result. $\bar{y}$ is the mean overall experimental data points.

### Reporting summary.

Further information on research design is available in the Nature Research Reporting Summary linked to this article.

## Data availability

All modeling results are available as CSV files from zenodo with the identifier https://doi.org/10.5281/zenodo.7084922[62].

## Code availability

The Python source code for the model, including scripts for deriving all results, are available on zenodo with the identifier https://doi.org/10.5281/zenodo.5643429[56].

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

## Acknowledgements

This work was supported by the European Research Council (ERC) under the European Union's Horizon 2020 Research and Innovation Programme grant agreement No. 716800 (ERC Starting Grant to M.D.)

## Author contributions

J.E. and M.D. designed the research; J.E. conducted calculations, analyzed the data, and produced figures; J.E. and M.D wrote the paper.

## Funding

## Competing interests

The authors declare no competing interests.
