## [Peer Review File · Communications Biology]

Reviewers' comments:

Reviewer #1 (Remarks to the Author):

This manuscript develops a new model of binaural (i.e., two-eared) hearing and tests the model's performance on a battery of prior human psychophysical data. For over 7 decades, the prevailing theory of binaural processing based on the interaural time difference (ITD, or equivalently, the interaural phase difference for a known frequency) cue to sound source location was the so-called Jeffress model. This model postulated three things: 1) phase locking of peripheral neuron spiking to the stimulus at each ear, 2) physical delay lines (longer axon path lengths) that compensate for the external acoustic ITD, and 3) coincidence detecting neurons that receive input from both ears and that fire maximally when action potentials arrive coincidentally. Finally, Jeffress postulated that there is a population of coincidence detectors that are differentially sensitive to a range of ITDs. Thus, a time-based cue, ITD, is converted to a place code, in essence a topographical mapping of ITD. The Jeffress neural architecture has been demonstrated in the anatomy and physiology of the barn owl brainstem. In the mammalian brainstem, however, phase locking and coincidence detecting neurons have been documented, but delay lines in the form of physically longer axons from the two ears has not been found. And thus, nothing approaching a topographical map of ITD has been found in the brainstem. More recently it has been proposed that the mammalian binaural system for ITD/IPD coding must be different than the classical Jeffress model. A two-channel hemispheric model of ITD/IPD coding has gathered support.

The motivation for this manuscript is that while the Jeffress model has had good success at characterizing a large battery of prior psychophysical data on tone-in-noise masking, two-channel models have not performed well. This paper develops a new model based on a mathematical model of two-channel IPD coding. The model is quite simple and elegant. It essentially captures the correlation of signals at the two ears and additionally a running computation of the IPD. The model is applied to eight different datasets and does a remarkable job of describing the results, with 98% of the variance explained. Even using a single set of parameters for the model instead of optimizing parameters for each dataset, the model explains 93% of the total variance.

The paper is well written and concise. The results support the conclusions. And the manuscript makes an important contribution and advancement to the field.

I have only a few suggestions.

First, in the introduction it is stated that the mammalian brainstem is vastly different than that of the Jeffress model or the barn owl. However, really it is only that delay lines are missing in mammals, and thus no topographical map of ITD as prescribed by Jeffress. But it is precisely this mapping that prior models of binaural unmasking have leveraged to great success. As stated by the authors, while the Jeffress-based models do well, this is not the architecture of the mammalian brainstem! The authors might make these points in the introduction.

One of the most difficult binaural unmasking datasets to model even with Jeffress are 'frozen' noise, such as the experiments of Gikley, Colburn and Carney to name just a few. Jeffress-type models can describe average performance over all tokens, but fail to predict performance for individual noise tokens. Sample to sample variation in the noise tokens from trial to trial was mentioned on line 288 of the manuscript as a difficult situation for the present model to handle. Modeling frozen noise binaural unmasking is beyond the scope of the present manuscript, but it might be mentioned as a difficult data set historically to model. The present model might also have difficulty describing these data.

Finally, beginning on line 216 there is a discussion about the necessity of a form of normalization for the correlation and how this might occur physiologically. In addition to the methods suggested, it should be noted that there are also brainstem neurons that act as anti-coincidence detectors, in the

lateral superior olive. These neurons are just as sensitive to ITD/IPD as the MSO neurons (see Tollin and Yin, 2005, Journal of Neuroscience). As a stimulus is decorrelated, MSO neurons reduce their responses but LSO neurons increase response. A comparison of the MSO and LSO responses is a proxy for interaural correlation. There are neurons at the next stage, the inferior colliculus, that receive inputs from both MSO and LSO neurons (Shackleton et al 2000, Hearing Research).

Reviewer #2 (Remarks to the Author):

The authors propose a new mathematical formulation for the two-channel model of binaural sound localization in mammals: a complex-valued correlation coefficient γ that contains the two channels as its real and imaginary parts. The new formulation allows to explain binaural unmasking and other psychoacoustic phenomena beyond spatial sound localization.

The manuscript is well written and the Introduction is succinct and clear, with a nice review of the field. The results are clearly presented and the figures are of very high quality. The results show impressive fitting of many different works in the literature. The Discussion also includes some limitations of the proposed model, although some of them could be improved (see below). The manuscript seems to be already reviewed, as it is very good and for instance has a very low number of typos.

In summary, I suggest publication after the following minor suggestions are addressed.

Minor suggestions:

- Please add a summary of the two-channel model in the Introduction (the Jeffress model is briefly described, yet the two-channel model isn't).

- Line 298: Equation (4) uses $S_{\{uu\}}$ and $S_{\{zz\}}$ but in the next line it changes to $S_{\{ll\}}$ and $S_{\{rr\}}$. Please correct

- Line 302: "and of"  "and"

- Lines 245-265: The description of the first described limitation (asymmetric IPD-rate functions) is very messy and it doesn't clearly state the limitation, as opposed to the subsequent limitations which are very clearly stated (line 267 "In the $N_0 S_0$ condition shown in Fig. 2h, the model deviates significantly from the data [...] This effect cannot be accounted for by the current model implementation", line 271 "The current model also neglects peripheral processing apart from bandpass filtering. Without this pre-processing, the model cannot account for effects that are", line 274 "Another limitation of the current implementation is its focus on a single frequency channel centered at 500 Hz"). Please clearly state the limitation.

Reviewer #3 (Remarks to the Author):

Encke and Dietz describe a simple and elegant model of binaural unmasking, that is quite mathematical in nature, but whose simple approach is inspired by neurophysiological data regarding sound localisation in small mammals. This data suggests that Jeffress' (1948) model for the detection of interaural time delays is quite overspecified. In Jeffress' model there is an array of cells in the brainstem that is innervated by both ears via axons that vary systematically in length. The variation in length creates differential conduction times from the two ears and so action potentials arriving simultaneously at a given neuron will encode an external sound with a particular interaural delay. The putative cells fire when such coincidences occur. The neurophysiology suggests, however, that there is not a finely tuned array of these cells, but two populations that are broadly tuned to just two different

interaural delays; intermediate delays can thus be coded by the relative activation of these two populations. The present paper extends this idea to the phenomenon of binaural unmasking. It further simplifies it through a mathematical abstraction; because the tuning of the two populations of cells is at approximately ± 45 degrees in each frequency band, the two channels appear to be in quadrature phase and so can efficiently encode all possible phase differences. The model is therefore reduced to mathematically deriving a phase and a coherence value in each band (the coherence can be thought of as the maximum of the cross-correlation function, but is defined slightly differently in footnote 1). Although it is clear where the equivalent phase would be derived physiologically, I was less clear to me where the coherence would come from in the brain using only the activations the two cell populations. A clearer statement is needed there.

The model includes an explicit integration of monaural and binaural cues using signal detection theory, which is nice to see.

A wide range of data is successfully fitted using the model, as shown in Figure 2. I would support the use of individually fitted model parameters for each dataset, because there are substantial quantitative differences between studies for the same conditions, which prevents a common set of parameters from producing a good fit across data sets. The choice of studies seems very appropriate, but they are all using 500 Hz as the signal frequency and in my experience getting correct predictions across frequency is one of the hardest things to do in this area. Using broadband noise, the binaural unmasking effect (the "BMLD") decreases with increasing frequency, but asymptotes to 2-3 dB above 1500 Hz. As I understand it, this mathematical model would behave almost identically at different frequencies. The authors remark that "the model cannot account for effects that are associated with the periphery" (line 272) which appears to be a slightly oblique reference to this problem, because most models in this area tackle the effect of frequency using a model of peripheral transduction. Such models reduce, with increasing frequency, the encoding of the temporal fine structure of the waveform, but conserve the encoding of the waveform envelope. I thought this discussion could be more explicit.

Another problem with the purely mathematical treatment is that the model uses phase information directly. As a consequence, it makes identical predictions for the conditions known as NoSpi and NpiSo, for which the interaural phases of the noise and the signal both differ by π radians. These conditions consistently differ empirically, with NoSpi giving a larger effect. The authors address this issue in the Discussion (from line 245) by invoking the asymmetric shape of the observed IPD rate functions (illustrated in Fig. 4a). Again, a more detailed physiological implementation is invoked as needed to capture all features of the data, and, again, I wonder about the effect of frequency. The difference between NoSpi and NpiSo is quite small at 500 Hz, but it grows substantially at lower frequencies. I would like to know how the rate function shape would predict this effect. Would it require greater asymmetry at lower frequencies, or would the same thing have a larger effect on predictions?

Writing

on lines 259 and 261 the word "then" appears to be used in place of "than"

Authors' Response to Reviews of

A hemispheric two-channel code accounts for binaural unmasking in humans.

Jörg Encke, Mathias Dietz

Communications Biology, COMMSBIO-22-1727-T

RC: Reviewers' Comment, AR: Authors' Response, □ Manuscript Text

1. Reviewer #1

1.1. General comment

RC: *This manuscript develops a new model of binaural (i.e., two-eared) hearing and tests the model's performance on a battery of prior human psychophysical data. For over 7 decades, the prevailing theory of binaural processing based on the interaural time difference (ITD, or equivalently, the interaural phase difference for a known frequency) cue to sound source location was the so-called Jeffress model. This model postulated three things: 1) phase locking of peripheral neuron spiking to the stimulus at each ear, 2) physical delay lines (longer axon path lengths) that compensate for the external acoustic ITD, and 3) coincidence detecting neurons that receive input from both ears and that fire maximally when action potentials arrive coincidentally. Finally, Jeffress postulated that there is a population of coincidence detectors that are differentially sensitive to a range of ITDs. Thus, a time-based cue, ITD, is converted to a place code, in essence a topographical mapping of ITD. The Jeffress neural architecture has been demonstrated in the anatomy and physiology of the barn owl brainstem. In the mammalian brainstem, however, phase locking and coincidence detecting neurons have been documented, but delay lines in the form of physically longer axons from the two ears has not been found. And thus, nothing approaching a topographical map of ITD has been found in the brainstem. More recently it has been proposed that the mammalian binaural system for ITD/IPD coding must be different than the classical Jeffress model. A two-channel hemispheric model of ITD/IPD coding has gathered support. The motivation for this manuscript is that while the Jeffress model has had good success at characterizing a large battery of prior psychophysical data on tone-in-noise masking, two-channel models have not performed well. This paper develops a new model based on a mathematical model of two-channel IPD coding. The model is quite simple and elegant. It essentially captures the correlation of signals at the two ears and additionally a running computation of the IPD. The model is applied to eight different datasets and does a remarkable job of describing the results, with 98% of the variance explained. Even using a single set of parameters for the model instead of optimizing parameters for each dataset, the model explains 93% of the total variance. The paper is well written and concise. The results support the conclusions. And the manuscript makes an important contribution and advancement to the field.*

AR: We thank Reviewer 1 for this positive evaluation of our manuscript. We have uploaded a version of the manuscript where all changes to the previous version are indicated. We implement the changes as suggested (see below). Line numbers in this document refer to those in the pdf with tracked changes.

1.2. Introduction

RC: *First, in the introduction it is stated that the mammalian brainstem is vastly different than that of the Jeffress model or the barn owl. However, really it is only that delay lines are missing in mammals, and thus no topographical map of ITD as prescribed by Jeffress. But it is precisely this mapping that prior models of binaural unmasking have leveraged to great success. As stated by the authors, while the Jeffress-based models do well, this is not the architecture of the mammalian brainstem! The authors might make these points in the introduction.*

AR: Yes, we agree with this assessment, and it was a point we tried to make in the original manuscript. We have slightly rewritten the respective parts to clarify this point. (L30–43)

[...] In mammals, however, no such structure has been found. Instead of a nearly frequency-independent distribution of best-delays centered around $\tau = 0$ as ideal for the Jeffress-model [1], studies found the best delay of neurons in each hemisphere of the brain to be centered around $1/8^{\text{th}}$ of the cycle duration [2–4] (See visualization in Fig. 1c). Mammals thus seem to lack the topographical map of ITDs, as postulated by Jeffress. These findings resulted in the formulation of an alternative coding hypothesis: The two-channel model. Instead of the large number of ~~differently-systematically~~ tuned coincidence detectors used by the Jeffress model, this model relies on the activity within only two broad hemispheric channels [2]. Instead of the Jeffress place code, the ITD is encoded by the relative firing rate change within both channels. The two-channel code thus represents a rate code. The two-channel approach has been incorporated into several quantitative models dealing with various aspects of binaural hearing [5–8], but between them, these models still lack the predictive power of cross-correlation based approaches. As a consequence, and despite ~~their disagreement with mammalian physiology~~ the apparent lack of systematic delay lines in mammals, Jeffress-type models are still widely used to account for experimental data in humans, especially when dealing with phenomena beyond sound localization [9, 10]. [...]

1.3. Frozen Noise

RC: *One of the most difficult binaural unmasking datasets to model even with Jeffress are ‘frozen’ noise, such as the experiments of Gikley, Colburn and Carney to name just a few. Jeffress-type models can describe average performance over all tokens, but fail to predict performance for individual noise tokens. Sample to sample variation in the noise tokens from trial to trial was mentioned on line 288 of the manuscript as a difficult situation for the present model to handle. Modeling frozen noise binaural unmasking is beyond the scope of the present manuscript, but it might be mentioned as a difficult data set historically to model. The present model might also have difficulty describing these data.*

AR: Yes, we agree with the reviewer, and we have added a short paragraph that points out this potential limitation in the manuscript (L295–298)

[...] signal waveforms should account for this phenomenon. Another data set that has historically been difficult to account for, employs binaural unmasking in reproducible noise token [11, 12]. These experiments tested tone-in-noise detection using the same reproducible token across subjects. Like other models based on the average stimulus, we expect the presented model only to explain the average performance but not the variability between individual tokens. [...]

1.4. Normalization

RC: *Finally, beginning on line 216 there is a discussion about the necessity of a form of normalization for the correlation and how this might occur physiologically. In addition to the methods suggested, it should be noted that there are also brainstem neurons that act as anti-coincidence detectors, in the lateral superior olive. These neurons are just as sensitive to ITD/IPD as the MSO neurons (see Tollin and Yin, 2005, Journal of Neuroscience). As a stimulus is decorrelated, MSO neurons reduce their responses but LSO neurons increase response. A comparison of the MSO and LSO responses is a proxy for interaural correlation. There are neurons at the next stage, the inferior colliculus, that receive inputs from both MSO and LSO neurons (Shackleton et al 2000, Hearing Research).*

AR: Thank you for pointing out this alternative normalization method. We have added it to the respective section (L226–229)

[...] ~~An alternative~~ Alternatively, normalization could also be based on the activity in “anti-coincidence” detecting neurons such as those found in the lateral superior olive [13]. The firing rate in these neurons behaves inversely to those that act as coincident detectors and increases with decreasing correlation. Comparison between anti-coincidence and coincidence detecting neurons could thus be used for normalization. A third method [...]

2. Reviewer #2

3. General Comment

RC: *The authors propose a new mathematical formulation for the two-channel model of binaural sound localization in mammals: a complex-valued correlation coefficient γ that contains the two channels as its real and imaginary parts. The new formulation allows to explain binaural unmasking and other psychoacoustic phenomena beyond spatial sound localization. The manuscript is well written and the Introduction is succinct and clear, with a nice review of the field. The results are clearly presented and the figures are of very high quality. The results show impressive fitting of many different works in the literature. The Discussion also includes some limitations of the proposed model, although some of them could be improved (see below). The manuscript seems to be already reviewed, as it is very good and for instance has a very low number of typos. In summary, I suggest publication after the following minor suggestions are addressed.*

AR: Thank you for this very positive review! We have addressed your suggestions as discussed below. Line numbers in this document refer to those in the pdf with tracked changes.

3.1. Summary

RC: *Please add a summary of the two-channel model in the Introduction (the Jeffress model is briefly described, yet the two-channel model isn't).*

AR: Thank you for pointing this out. We have extended the description of the two-channel model. (See our response to comment 1.2)

3.2. Line 298

RC: Equation (4) uses S_{uu} and S_{zz} but in the next line it changes to S_{ll} and S_{rr} . Please correct

AR: Fixed

3.3. Line 302

RC: "and of" -> "and"

AR: Fixed

3.4. lines 245-265

RC: *The description of the first described limitation (asymmetric IPD-rate functions) is very messy and it doesn't clearly state the limitation, as opposed to the subsequent limitations which are very clearly stated line 267: "In the N_0S_0 condition shown in Fig. 2h, the model deviates significantly from the data [...] This effect cannot be accounted for by the current model implementation", line 271 "The current model also neglects peripheral processing apart from bandpass filtering. Without this pre-processing, the model cannot account for effects that are", line 274 "Another limitation of the current implementation is its focus on a single frequency channel centered at 500 Hz". Please clearly state the limitation.*

AR: We agree and have added the following sentence to state the limitation clearly: (L266–267)

[. . .] The experiment shown in Fig. 2h revealed differences between the threshold SNRs for in-phase tones in anti-phasic noise ($N_\pi S_0$) and anti-phasic tones in in-phase noise ($N_0 S_\pi$): thresholds in the ($N_\pi S_0$) condition were about 4 dB higher than for $N_0 S_\pi$. The same trend can be seen in the data shown in Fig. ??b and is consistent across other studies [14, 15]. ~~For our~~ This 4 dB difference between the $N_\pi S_0$ and the $N_0 S_\pi$ condition can not be accounted for by the presented model. In the model, the two conditions differ only in their mean IPD (that is, in the argument of the correlation coefficient $\arg(\gamma)$), which is zero for the $N_0 S_\pi$ condition and π for $N_\pi S_0$. [. . .]

4. Reviewer #3

5. General Comment

RC: *Encke and Dietz describe a simple and elegant model of binaural unmasking, that is quite mathematical in nature, but whose simple approach is inspired by neurophysiological data regarding sound localisation in small mammals. This data suggests that Jeffress' (1948) model for the detection of interaural time delays is quite overspecified. In Jeffress' model there is an array of cells in the brainstem that is innervated by both ears via axons that vary systematically in length. The variation in length creates differential conduction times from the two ears and so action potentials arriving simultaneously at a given neuron will encode an external sound with a particular interaural delay. The putative cells fire when such coincidences occur. The neurophysiology suggests, however, that there is not a finely tuned array of these cells, but two populations that are broadly tuned to just two different interaural delays; intermediate delays can thus be coded by the relative activation of these two populations. The present paper extends this idea to the phenomenon of binaural unmasking. It further simplifies it through a mathematical abstraction; because the tuning of the two populations of cells is at approximately +/-45 degrees in each frequency band, the*

two channels appear to be in quadrature phase and so can efficiently encode all possible phase differences. The model is therefore reduced to mathematically deriving a phase and a coherence value in each band (the coherence can be thought of as the maximum of the cross-correlation function, but is defined slightly differently in footnote 1).

AR: We thank the reviewer for the positive evaluation of the manuscript and the insightful comments. We have addressed all comments (please see responses below). Line numbers in this document refer to those in the pdf with tracked changes.

5.1. Comment 1

RC: *Although it is clear where the equivalent phase would be derived physiologically, I was less clear to me where the coherence would come from in the brain using only the activations the two cell populations. A clearer statement is needed there.*

AR: Thank you for pointing out that we weren't clear enough when addressing this. We have added a paragraph discussing the neuronal computations necessary to derive coherence from the two channels. We would, however, prefer not to speculate about the exact neuronal circuitry or where the circuitry is located. (L235–244)

[...] This IPD-fluctuation code would have the benefit of only requiring information about a single quantity – the IPD – ~~but would also require neurons.~~

The neuronal substrate necessary to extract γ based on the two channels would depend on the underlying mechanism. Representing γ directly based on the values of coherence and mean-IPD would require a long-term integration mechanism with two subsequent processing stages implementing neuronal equivalents to calculating modulus and argument. If γ would instead be represented indirectly via IPD fluctuations, neurons would have to be fast enough to follow ~~the instantaneous changes in IPD caused by a reduction in coherence~~ these fluctuations. Neurons that do just this have indeed been described: IPD-sensitive neurons can encode the fast fluctuations ~~by~~ by means of fast changes in their response rate [16–18]. This second mechanism would also be in-line with a recent neuro-imaging study which reported an elevated cortical load when subjects were presented with low-coherence stimuli [19]. The increased load could result from the brain having to deal with localizing a sound source based on increasingly fluctuating IPDs. [...]

5.2. Comment 2

RC: *A wide range of data is successfully fitted using the model, as shown in Figure 2. I would support the use of individually fitted model parameters for each dataset, because there are substantial quantitative differences between studies for the same conditions, which prevents a common set of parameters from producing a good fit across data sets. The choice of studies seems very appropriate, but they are all using 500 Hz as the signal frequency and in my experience getting correct predictions across frequency is one of the hardest things to do in this area. Using broadband noise, the binaural unmasking effect (the "BMLD") decreases with increasing frequency, but asymptotes to 2-3 dB above 1500 Hz. As I understand it, this mathematical model would behave almost identically at different frequencies. The authors remark that "the model cannot account for effects that are associated with the periphery" (line 272) which appears to be a slightly oblique reference to this problem, because most models in this area tackle the effect of frequency using a model of peripheral transduction. Such models reduce, with increasing frequency, the encoding of the temporal fine structure of the waveform, but conserve the encoding of the waveform*

envelope. I thought this discussion could be more explicit.

AR: Yes, we agree that this discussion could be more explicit and added a corresponding paragraph. (L300–307)

[...] The lack of realistic peripheral processing also limits the ability of the model to account for differences in binaural detection with frequency. IPD and thus ITD sensitivity rely on the auditory nerves' activity to lock onto the stimulus phase (so-called phase-locking). Phase locking declines with increasing frequency [20] which is one of the reasons why sensitivity to ITDs and binaural detection thresholds worsen with increasing frequency [14, 21]. The loss of phase-locking could be implemented by a low-pass filter included in a model of peripheral transduction. This filter would remove phase information at higher frequencies while keeping the waveform envelope. Extending the current model with a more detailed periphery should ~~help account for these data sets.~~

thus help to account for the effects associated with the periphery and the reduction in phase sensitivity at higher frequencies. The current implementation also uses only one single frequency channel. ~~Another limitation of the current implementation is its focus on a single frequency channel centered at 500 Hz. With different parametrization, the model can be reasonably expected to account for experiments at different frequencies.~~ Other experiments, however, such as those employing spectrally complex maskers or maskers constructed from two noise sources with different ITDs, might require a multi-channel implementation of the model. The success of this kind of model extension has recently been demonstrated [22]. [...]

5.3. Comment 3

RC: *Another problem with the purely mathematical treatment is that the model uses phase information directly. As a consequence, it makes identical predictions for the conditions known as NoSpi and NpiSo, for which the interaural phases of the noise and the signal both differ by π radians. These conditions consistently differ empirically, with NoSpi giving a larger effect. The authors address this issue in the Discussion (from line 245) by invoking the asymmetric shape of the observed IPD rate functions (illustrated in Fig. 4a). Again, a more detailed physiological implementation is invoked as needed to capture all features of the data, and, again, I wonder about the effect of frequency. The difference between NoSpi and NpiSo is quite small at 500 Hz, but it grows substantially at lower frequencies. I would like to know how the rate function shape would predict this effect. Would it require greater asymmetry at lower frequencies, or would the same thing have a larger effect on predictions?*

AR: It would indeed be necessary to change the shape of IPD-rate functions with frequency to capture the frequency-dependent differences between the two conditions. This assumption, however, is not unreasonable as physiological recordings also show increasingly harmonic ITD-rate functions. We have added a paragraph with this discussion to the manuscript. (L285–289)

[...] Differences between N_0S_π and $N_\pi S_0$ are larger at low frequencies and get smaller with increasing frequency [14]. If these differences result from non-sinusoidal IPD-rate functions, then one would also expect the IPD-rate functions to become more sinusoidal with increasing frequency. This assumption is indeed supported by physiological data [23, 24] which shows increasingly harmonic ITD-rate functions with increasing frequency. [...]

5.4. lines 259 and 261

RC: *the word "then" appears to be used in place of "than"*

AR: Fixed

References

- [1] RM Stern, GD Shear, Lateralization and detection of low-frequency binaural stimuli: Effects of distribution of internal delay. *The J. Acoust. Soc. Am.* **100**, 2278–2288 (1996). .
- [2] D McAlpine, D Jiang, AR Palmer, A neural code for low-frequency sound localization in mammals. *Nat. Neurosci.* **4**, 396–401 (2001). .
- [3] PX Joris, BV de Sande, DH Louage, M van der Heijden, Binaural and cochlear disparities. *Proc. Natl. Acad. Sci.* **103**, 12917–12922 (2006). .
- [4] T Marquardt, D McAlpine, *A π -Limit for Coding ITDs: Implications for Binaural Models*, eds. B Kollmeier, et al. (Springer, Berlin, Heidelberg), pp. 407–416 (2007).
- [5] M Dietz, SD Ewert, V Hohmann, B Kollmeier, Coding of temporally fluctuating interaural timing disparities in a binaural processing model based on phase differences. *Brain Res.* **1220**, 234–245 (2008). .
- [6] M Takanen, O Santala, V Pulkki, Visualization of functional count-comparison-based binaural auditory model output. *Hear. Res.* **309**, 147–163 (2014). .
- [7] J Encke, W Hemmert, Extraction of inter-aural time differences using a spiking neuron network model of the medial superior olive. *Front. Neurosci.* **12** (2018). .
- [8] J Bouse, V Vencovský, F Rund, P Marsalek, Functional rate-code models of the auditory brainstem for predicting lateralization and discrimination data of human binaural perception. *The J. Acoust. Soc. Am.* **145**, 1–15 (2019). .
- [9] LR Bernstein, C Trahiotis, Binaural detection as a joint function of masker bandwidth, masker interaural correlation, and interaural time delay: Empirical data and modeling. *The J. Acoust. Soc. Am.* **148**, 3481–3488 (2020). .
- [10] LR Bernstein, C Trahiotis, Accounting for binaural detection as a function of masker interaural correlation: Effects of center frequency and bandwidth. *The J. Acoust. Soc. Am.* **136**, 3211–3220 (2014). .
- [11] J Mao, LH Carney, Binaural detection with narrowband and wideband reproducible noise maskers. iv. models using interaural time, level, and envelope differences. *The J. Acoust. Soc. Am.* **135**, 824–837 (2014). .
- [12] SA Davidson, RH Gilkey, HS Colburn, LH Carney, An evaluation of models for diotic and dichotic detection in reproducible noises. *The J. Acoust. Soc. Am.* **126**, 1906 (2009). .
- [13] DJ Tollin, TCT Yin, Interaural phase and level difference sensitivity in low-frequency neurons in the lateral superior olive. *The J. neuroscience : official journal Soc. for Neurosci.* **25**, 10648–10657 (2005). .
- [14] IJ Hirsh, The influence of interaural phase on interaural summation and inhibition. *The J. Acoust. Soc. Am.* **20**, 536–544 (1948). .

- [15] DE Robinson, LA Jeffress, Effect of varying the interaural noise correlation on the detectability of tonal signals. *The J. Acoust. Soc. Am.* **35**, 1947–1952 (1963). .
- [16] PX Joris, B van de Sande, A Recio-Spinoso, M van der Heijden, Auditory midbrain and nerve responses to sinusoidal variations in interaural correlation. *J. Neurosci.* **26**, 279–289 (2006). .
- [17] PX Joris, Neural binaural sensitivity at high sound speeds: Single cell responses in cat midbrain to fast-changing interaural time differences of broadband sounds. *The J. Acoust. Soc. Am.* **145**, EL45–EL51 (2019). .
- [18] I Siveke, SD Ewert, B Grothe, L Wiegrebe, Psychophysical and physiological evidence for fast binaural processing. *J. Neurosci.* **28**, 2043–2052 (2008). .
- [19] R Luke, H Innes-Brown, JA Undurraga, D McAlpine, Human cortical processing of interaural coherence. *iScience* **25**, 104181 (2022). .
- [20] DH Johnson, The relationship between spike rate and synchrony in responses of auditory-nerve fibers to single tones. *The J. Acoust. Soc. Am.* **68**, 1115–1122 (1980). .
- [21] A Brughera, L Dunai, WM Hartmann, Human interaural time difference thresholds for sine tones: The high-frequency limit. *J. Acoust. Soc. Am.* **133**, 2839 (2013). .
- [22] B Eurich, J Encke, SD Ewert, M Dietz, Interaural coherence across frequency channels accounts for binaural detection in complex maskers (arXiv 2110.02695) (2021).
- [23] TC Yin, S Kuwada, Binaural interaction in low-frequency neurons in inferior colliculus of the cat. iii. effects of changing frequency. *J. Neurophysiol.* **50**, 1020–1042 (1983).
- [24] D McAlpine, Creating a sense of auditory space. *The J. Physiol.* **566**, 21–28 (2005). .

REVIEWERS' COMMENTS:

Reviewer #1 (Remarks to the Author):

The authors satisfactorily addressed all concerns.

Reviewer #3 (Remarks to the Author):

I am content with the authors' revisions and recommend publication.